# Synthesis and Biological Evaluation of 2,3,4-Triaryl-1,2,4-oxadiazol-5-ones as p38 MAPK Inhibitors

**DOI:** 10.3390/molecules26061745

**Published:** 2021-03-20

**Authors:** Roberto Romeo, Salvatore V. Giofrè, Maria A. Chiacchio, Lucia Veltri, Consuelo Celesti, Daniela Iannazzo

**Affiliations:** 1Dipartimento di ScienzeChimiche, Biologiche, Farmaceutiche ed Ambientali, Università di Messina, Via S.S. Annunziata, 98168 Messina, Italy; sgiofre@unime.it; 2DipartimentoScienze del Farmaco, Università di Catania, Viale A. Doria 6, 95125 Catania, Italy; ma.chiacchio@unict.it; 3Dipartimento di Chimica e TecnologieChimiche, Universitàdella Calabria, Via P. Bucci 12/C, 87036 Arcavacata di Rende, Italy; lucia.veltri@unical.it; 4Dipartimento di Ingegneria, Università di Messina, Contrada Di Dio, 98166 Messina, Italy; ccelesti@unime.it (C.C.); diannazzo@unime.it (D.I.)

**Keywords:** stilbene analogs, 1,2,4-oxazolidinyl-5-ones, p38 MAPK inhibitors

## Abstract

A series of azastilbene derivatives, characterized by the presence of the 1,2,4-oxadiazole-5-one system as a linker of the two aromatic rings of stilbenes, have been prepared as novel potential inhibitors of p38 MAPK. Biological assays indicated that some of the synthesized compounds are endowed with good inhibitory activity towards the kinase. Molecular modeling data support the biological results showing that the designed compounds possess a reasonable binding mode in the ATP binding pocket of p38α kinase with a good binding affinity.

## 1. Introduction

The stilbene scaffold is a basic element for several biologically active natural and synthetic compounds which have demonstrated promising activity in the area of medicinal chemistry, targeting a wide variety of intracellular pathways [1,2,3,4,5,6,7]. Stilbene derivatives are of significant interest for drug research and development because of their potential in a wide variety of pathophysiological conditions, for their antioxidant, anti-inflammatory, antiaging, antidiabetic, antiviral, neuroprotective, cardioprotective, and anticancer properties [8,9,10,11,12,13,14,15,16,17,18,19,20,21]. In order to improve the chemopreventive and/or therapeutic activities, as well as the bioavailability with respect to the parent compound, new stilbene derivatives have been designed, synthesized, and tested on multiple cellular targets.

A key structural factor for the biological activity is the presence of the double bond in a Z configuration: the olefinic bond forces the two aromatic rings to stay within an appropriate distance and gives the molecule the right dihedral angle to maximize the interaction with the target. The Z stilbene double bond can easily isomerize under the influence of heat, light, and protic media; therefore, many attempts have been addressed to modify the olefinic bridge in order to stabilize the configuration and to increase the biological effects of the compound [22,23]. In this context, the problem of the isomerization of the active *cis* double bond into an inactive form has been also achieved by the exploitation of heterocyclic rings in place of the ethene bridge [24,25,26]. These *cis*-locked analogs provide some interesting advantages such as the prevention of *cis* to *trans* isomerization, increased specificity since the *trans* conformation might be recognized by other cellular targets, and the presence of heterocyclic units that might improve the therapeutic potential of the synthesized potential drugs [22,23].

Three-, four-, five-, and six-membered rings have been exploited to replace the olefinic moiety. Among the five-membered rings, 2-cyclopenten-1-ones (Figure 1 (**1**)) [27], 2(5H)-furanones (Figure 1 (**2**)) [28], 1,3-oxazoles (Figure 1 (**3**)) [29] and 4,5-dihydroisoxazoles (Figure 1 (**4**)) [30], diaryloxazolones (Figure 1 (**5**)) [31], furazans (Figure 1 (**6**)) [32], imizadoles (Figure 1 (**7**)) [29], pyrazoles (Figure 1 (**8**)) [29], triazoles (Figure 1 (**9**)) [33], 2,3-dihydrothiophenes (Figure 1 (**10**)) [34], 4,5-disubstituted imidazoles (Figure 1(**11a** and **11b**) [29], and arylcoumarins (Figure 1 (**12**)) [35] have been reported. 2,3,8,8a-Tetrahydro-5(1H)-indolizinones (Figure 1 (**13**)) [36], benzene and pyridine (Figure 1 (**14**)) and (Figure 1 (**15**)) [30] were also used as six-membered rings; two reports have appeared with three-membered rings [37,38]. Among the four-membered rings, azetidone analogs (Figure 1 (**16**)) were prepared [39] (Figure 1).

Further optimization of the structural diversity of stilbenes to increase their potency and biological activity has been explored. In this context, several *cis*-locked stilbene-based compounds with vicinal 4-fluorophenyl/4-pyridine rings were designed, synthesized, and evaluated as inhibitors of p38α mitogen-activated protein kinase (MAPK). The mitogen-activated protein kinases (MAPKs) are essential regulators for signal transduction pathways, involved in the control of cell functions including proliferation, gene expression, differentiation, mitosis, cell survival, and apoptosis [40]. The p38 MAP kinases are responsive to stress stimuli, such as ultraviolet irradiation, heat shock, osmotic shock, and mechanical stress; the abnormal activity of P38 is involved in the production of cytokines (IL-1 and TNFα), which are implicated in chronic inflammatory diseases [41] of several tissues, that include neuronal [42,43,44], bone [45], lung [46], cardiac and skeletal muscle [47,48], red blood cells [49], and fetal tissues [50]. Consequently, the discovery of new molecules that can inhibit the p38 MAP kinases could afford an effective therapy for the treatment of these autoimmune diseases. Moreover, recently, the blockade of the p38α MAPK signaling pathway has been suggested to be a promising strategy for the treatment of several different cancers, such as breast cancer, colon cancer, and ovarian cancer [51,52,53,54,55].

Several selective stilbene-based p38 inhibitors, where the vicinal 4-fluorophenyl/pyridinyl motif is inserted into different ring scaffolds, have been developed, such as five-membered pyrazoles ((Figure 2 (**1**)) [56,57], isoxazoles (Figure 2 (**2**)) [58,59], triazoles (Figure 2 (**3**)) [60,61,62,63], and six-membered quinolinones (Figure 2 (**4**)) [64,65], pyrimidines (Figure 2 (**5**)) [66], and chromones (Figure 2 (**6**)) [67]. Many of these compounds have been shown to be highly potent and able to inhibit the p38α kinase activity at nanomolar concentrations (Figure 2).

Following the strategy based on the use of heterocyclic systems to replace the ethene bridge of biologically active stilbenes, we have explored if it is possible the substitution of the Z double bond with a conformationally restricted cyclic C-N bond and still keep a strong biological effect. Here we present the design, synthesis, and biological evaluation of new *cis*-restricted azastilbenes (Figure 3), able to bind to the ATP-binding pocket of p38 MAP kinase, characterized by the presence of an oxadiazole ring, as a linker of the two aromatic rings of stilbenes, with the C=C double bond replaced by a conformationally restricted C-N bond.

## 2. Results and Discussion

### 2.1. Design

The binding of several different azastilbenes, characterized by the presence of a 4-fluorophenyl/pyridine motif, to the ATP pocket of the p38α kinase has been well described; in particular, the binding mode of SB 203,580 [68], a well-known pyridinylimidazole inhibitor, is reported in Figure 4 [57,66].

Crystallographic data indicate that the interaction of SB 203,580 withthe ATP pocket of the p38 kinase involves the formation of a hydrogen bonding by the pyridine nitrogen atom with the backbone NH of Met109 in the hinge region. Furthermore, the interaction with the enzyme is further stabilized by hydrophobic interactions originated by the 4-fluorophenyl ring of the inhibitor which occupies the “hydrophobic pocket I” (HPI). Additional ligand-p38R interactions can be detected such as hydrogen bonding toward the core system from Lys53 and Tyr35 π-π stacking with the phenyl system. The binding mode of SB 203,580 shows that the ligand is not interfering with “hydrophobic region II” [57,66]. Thus, optimization of inhibitors to address HRII, which could be accomplished successfully mainly by introducing lipophilic substituents in the core system, appears to be a possible valuable improvement in the design of new inhibitors.

According to these considerations, our design has been addressed to the construction of a new series of novel potential inhibitors of MAPK3, in which two vicinal aromatic systems at C-3 and N-2 are connected to a 1,2,4-oxadiazole-5-one system. The heterocyclic ring assures the required correct spatial assessment of two vicinal aryl groups, while the stabilizing hydrogen bond interactions to the hinge region and Lys53 of the enzyme pocket are mimed by the nitrogen atom of the pyridinyl unit at C-3 and by the oxygen atom of the carbonyl group, respectively (Figure 3). Moreover, important hydrophobic interactions can be exerted by the aryl groups present at position 2: the aromatic system was supposed to be orientated to afford further consistent lipophilic interactions to HRII and to provide options for further structural modifications.

The biological assays indicated that some of the synthesized compounds are endowed with a good inhibitory activity towards the p38α kinase. The synthesized 3,4-diaryl-1,2,4-oxadiazol-5-ones **3a**–**l** possess a reasonable binding mode in the ATP binding pocket of p38α kinase.

### 2.2. Synthesis of 2,4-Oxazolidinyl-5-Ones (Azastilbenes)

The synthetic approach towards the differently substituted 1,2,4-oxazolidinyl-5-ones (azastilbenes) **3a–l** exploits a recently reported route [69] based on the 1,3-dipolar cycloaddition of nitrones **1a**–**e** to isocyanates as dipolarophiles. Isocyanates **2a**–**d** were reacted at room temperature in dry acetone and in a nitrogen atmosphere with nitrones **1a**–**e**, obtained by the reaction of the corresponding aldehydes with N-methyl or N-aryl hydroxylamine [70,71,72,73], leading to the expected cycloadducts **3a–l** in good yields (Figure 5; Table 1).

The structure of the obtained cycloadducts has been assigned on the basis of spectroscopic data (Appendix A). The ^1^H-NMR spectra of the compounds show the presence of the diagnostic methine proton at C-3, whose resonance, as previously reported [38], is in the range 5.80–6.10 ppm, typical of an NCH(Ph)N system, as in compound **3a–l**. Furthermore, the methane ^13^C resonates as a doublet at 79–80 ppm, values lower than those of the ^13^C possible which are expected at higher fields (85–90 ppm).

As further support to the assigned structure, the presence in the MS spectra of the diagnostic peak at M-44, resulting from the loss of CO_2_ from the molecular ion, is only amenable to cycloadducts **3a–l**.

As previously reported [69], the reaction showed a complete sito- and regioselectivity: under the adopted experimental conditions, only 1,2,4-oxadiazolidin-5-ones **3a–l** have been obtained as exclusive cycloadducts with no traces of the alternative cycloadducts (Figure 5). The observed sitoselectivity has been recently supported by DFT calculations which indicated that the cycloaddition takes place in dichloromethane through a stepwise process, where the formation of compounds **3a**–**l**, kinetically and thermodynamically favored, is exergonic for 13.0 kcal/mol [74].

### 2.3. Inhibition of p38α MAPK for Compounds ***3a**–**l***

The inhibitory potencies towards p38α MAPK of the compounds **3a**–**l** were evaluated in a non radioactive immunosorbent p38R MAPK enzyme assay, in which ATP competes with the inhibitor for the same binding pocket in the catalytic site of p38 α MAPK and compared with that exerted by the known p38 MAPK inhibitor SB203580 [41]. According to the effectiveness of the inhibitor and the resulting p38 α kinase activity, the phosphorylation reaction of the activating transcription factor 2 (ATF-2), the endogenous protein substrate of p38 MAPK, is suppressed to a greater or lesser extent so that the amount of phosphorylated ATF-2 inversely correlates with the inhibitory activity of the p38 α MAPK inhibitor [75].

The obtained results (Table 2) highlight the crucial importance, in the binding to the p38α kinase, of the formation of the hydrogen bond to the hinge region with the nitrogen atom of the pyridine ring: azastilbenes 3g–l do not show any significant inhibition of the p38 activity (Table 1; entries 7–10). For these compounds, the possible stabilizing interactions of the aryl substituent at N-4 with the hydrophobic pocket HPI and the hydrogen bond between the carbonyl group of the 1,2,4-oxadiazole ring and Lys53 of p38 MAP kinase do not appear relevant. Analogously, the possible additional lipophilic interactions to HRII, exerted by the substituent present at N-2 (compounds **3g**, **3h**, **3i**), do not lead to the expected biological activity.

Compounds witha 4-fluoro-phenyl/pyridinyl motif (entries 2,5,7) are shown to be endowed with good inhibitory activity, with an IC_50_ in the range of 0.10–5.1 µM. These values are perfectly in agreement with that obtained from the reference compound SB203580 (IC_50_ = 0.3 µM), thus highlighting the importance of a 4-fluorophenyl/pyridine motif in the molecular binding with the ATP pocket of the p38α kinase. In these compounds, the pyridine unit at C-3 in the oxazolidinone ring is able to exert the critical determinant hydrogen-bonding interaction in the hinge region of the enzyme. The most active compounds, 3e (0.08 mM) and 3f (1.1 mM), are characterized, besides the presence of the 4-fluoro-phenyl/pyridinyl motif, by an additional hydrophobic unit at N-2: the comparison with compounds 3l, which possesses a Me moiety (instead of phenyl or benzyl analog 3i), indicating that a consistent lipophilic interaction to HRII region of the enzyme could be the reason for the improved binding affinity of these derivatives. The substitution of the fluorine atom of 3e with a nitro group, CF_3_ or Cl, did not lead to anincrease of the inhibition of p38α for compounds 3a, 3c, and 3d.

A possible binding mode for 3e, chosen as a model compound is shown in Figure 6. According to literature data [37], the 2D structure completely fits with the ATP site of p38α, showing a suitable orientation of the vicinal pyridine/fluorophenylpharmacophore with accepting H-bonds both from Met109 by pyridine nitrogen as well as from quite flexible Lys53 by the carbonyl group of the 1,2,4-oxadiazole system. The two aromatic residues clasp around gatekeeper residue Thr106 in p38R, while the aryl group at N-2 interacts favorably with the HPII region.

The possible binding mode of 3e suggests that several suitably substituted 1,2,4-oxadiazol-5-ones can mimic the binding mode of the known imidazole-based SB 203,580 inhibitor, i.e., the inhibitor hydrogen bonds to the hinge region and the phenyl ring can interact with the hydrophobic pocket. Moreover, the C=C double bond of the core ring system of the inhibitors witha 4-fluoro-phenyl/pyridinyl motif can be replaced with the simple C-N bond of the oxadiazole system and still obtain good inhibitory capacity.

## 3. Experimental Section

### 3.1. Materials and Methods

Solvents and reagents are commercial. HRMS were determined with an SQ Quantum XLS Triple Quadrupole GC-MS/MS. NMR spectra (^1^H-NMR at 300 and 500 MHz, ^13^C-NMR at 75 and 125 MHz) were recorded with Varian instruments and are reported in ppm relative to TMS. Merck silica gel 60-F254 precoated aluminum plates have been used for thin-layer chromatographic separations. Flash chromatography was performed on Merck silica gel (Merck KGaA, Darmstadt, Germany) (200–400 mesh). Preparative separations were carried out by a MPLC Büchi C-601 using Merck silica gel 0.040–0.063 mm.

### 3.2. General Procedure

Compounds were synthesized according to a recent report [38]. A solution of nitrone (1 mmol) in 3 mL of acetone wasadded to a solution of isocyanate (1 mmol) in 5 mL of anhydrous acetone. The reaction mixture was left under stirring for 5 h; after removal of the solvent by evaporation under vacuum, the obtained yellow solid was triturated with methanol (10 mL) for 2 h. After filtration, the solid was purified by silica gel column chromatography. The synthesis of compounds 3a–b has been previously reported [38].

*2-Phenyl-4-(4-trifluoromethyl)-3-(pyridin-4-yl)-1,2,4-oxadiazolidin-5-one* (**3c**). Eluent cyclohexane/ethyl acetate 3:1; yellow sticky oil (yield 68%); ν max/cm^−1^: 1730 (C=O). ^1^H-NMR (500 MHz, CDCl_3_): δ 8.14 (d, *J* = 9.0 Hz, 2H), 7.51 (d, *J* = 9.0 Hz, 2H), 7.37–7.30 (m, 5H), 7.16. (d, *J* = 8.0 Hz, 2H), 6.91 (d, *J* = 8.0 Hz, 2H), 5.87 (s, 1H). ^13^C-NMR (500 MHz, CDCl_3_): δ 153.3, 150.0, 149.8, 146.5, 142.4, 133.8, 132.1, 127.1, 126.5, 125.3, 124.2, 124.1, 116.6, 89.4. HRMS-EI (*m*/*z*) calcd for C_20_H_14_F_3_N_3_O_2_ 385.3462, found 385, 3470.

*2-Phenyl-4-(4-chlorophenyl)-3-(pyridin-4-yl)-1,2,4-oxadiazolidin-5-one* (**3d**). Eluent cyclohexane/ethyl acetate 3: 1; yellow sticky oil (yield 58%). ν max/cm^−1^: 1730 (C=O). ^1^H-NMR (500 MHz, CDCl_3_): δ 8.10 (d, *J* = 7.8, 2H), 7.48 (d, *J*=7.8, 2H), 7.40–7.30 (m, 4H), 7.25–6.97 (m, 5H), 5.73 (s, 1H). ^13^C-NMR (500 MHz, CDCl_3_): δ 153.3, 150.0, 149.8, 146.5, 137.2, 133.3, 129.0, 127.1, 126.6, 126.5, 124.2, 89.4. HRMS-EI (*m*/*z*) calcd for C_22_H_19_ClN_2_O_3_ C_19_H_14_ClN_3_O_2_ 351.5607, found 351.5612.

*2-Benzyl-4-(4-fluorophenyl)-3-(pyridin-4-yl)-1,2,4-oxadiazolidin-5-one* (**3e**). Eluent cyclohexane/ethyl acetate 3: 1; yellow sticky oil (yield 71%).ν max/cm^−1^: 1730 (C=O). ^1^H-NMR (500 MHz, CDCl_3_): δ 8.55 (d, *J* = 8.2 Hz, 2H), 7.80 (d, *J* = 7.5 Hz, 2H), 7.35–7.33 (m, 4H), 7.27–7.20 (m, 5H), 6,05 (s, 1H), 3.86 (s, 2H). ^13^C-NMR (500 MHz, CDCl_3_): δ 162.9, 153.3, 149.8, 137.7, 134.7, 128.5, 127.9, 127.0, 124.2, 123.2, 115.7, 86.9, 59.9. HRMS-EI (*m*/*z*) calcd for C_20_H_16_FN_3_O_2_ 344.1330, found 344.1341.

*2-Benzyl-4-(4-nitrophenyl)-3-(pyridin-4-yl)-1,2,4-oxadiazolidin-5-one* (**3f**). Eluent cyclohexane/ethyl acetate 4: 1; yellow sticky oil (yield 81%). ν max/cm^−1^: 1730 (C=O).^1^H-NMR (500 MHz, CDCl_3_):δ 8.30 (d, *J* = 7.5 Hz, 2H), 7.45–7.40 (m, 4H), 7.35–7.20 (m, 7H), 6.04 (s, 1H), 3.95 (s, 2H). ^13^C-NMR (500 MHz, CDCl_3_): δ 153.3, 149.8, 146.5, 145.2, 143.5, 137.7, 131.1, 128.5, 127.9, 127.0, 124.2, 124.1, 86.9, 59.9. HRMS-EI (*m*/*z*) calcd for C_20_H_16_N_4_O_4_ 376.1275, found 376.1282.

*2-Benzyl-4-(4-fluorophenyl)-3-phenyl-1,2,4-oxadiazolidin-5-one* (**3g**). Eluent cyclohexane/ethyl acetate 4: 1; yellow sticky oil (yield 68%). νmax/cm^−1^: 1730 (C=O). ^1^H-NMR (500 MHz, CDCl_3_): δ7.55 (d, *J*=7.0 Hz, 2H), 7.45 (d, *J*=7.8 Hz, 2H), 7.35–7.20 (m, 10H), 5.90 (s, 1H9, 3,75 (s, 2H). ^13^C-NMR (500 MHz, CDCl_3_): δ162.9, 153.3, 139.2, 137.5, 134.7, 128.5, 127.9, 127.0, 126.9, 123.2, 115.7, 86.9, 59.9.HRMS-EI (M7z) calcd for C_21_H_17_FN_2_O_2_ 347.1163, found347.1170.

*2-Benzyl-4-(4-nitrophenyl)-3-phenyl-1,2,4-oxadiazolidin-5-one* (**3h**).Eluent cyclohexane/ethyl acetate 4 1; yellow sticky oil (yield 70%). νmax/cm^−1^: 1730 (C=O). ^1^H-NMR (500 MHz, CDCl_3_): δ8.25 (d, *J*=8.0 Hz, 2H), 7.45 (d, *J*=7.5 Hz, 2H), 7.40 (d, *J*= 8.0 Hz, 2H), 7.35–7.25 (m, 8H), 6.10 (s, 1H), 3.80 (s, 2H). ^13^C-NMR (500 MHz, CDCl_3_): δ 153.3, 145.2, 143.5, 139.2, 137.7, 131.1, 128.5, 127.9, 127.0, 126.7, 124.1, 86.9, 59.9. HRMS-EI (*m*/*z*) calcd for C_21_H_16_N_3_O_4_ 335.1063, found 335.1070.

*2-Benzyl-4-(4-fluorophenyl)-3-methyl-1,2,4-oxadiazolidin-5-one* (**3i**). Eluent cyclohexane/ethyl acetate 4: 1; yellow sticky oil (yield 70%). νmax/cm^−1^: 1730 (C=O).^1^H-NMR (500 MHz, CDCl_3_): δ 7.8 (q, *J*=7.2 Hz, 2H), 7.3–7.8 (m, 7H), 4.95 (q, *J*= 9.8 Hz, 1H), 3.85 (s, 2H), 1.30 (d, *J*=9.8 Hz, 3H). ^13^C-NMR (500 MHz, CDCl_3_): δ 162.9, 153.0, 137.7, 134.7, 128.5, 127.9, 127.0, 123.2, 115.7, 78.7, 60.0, 19.6. HRMS-EI (*m*/*z*) calcd for C_16_H_15_ FN_2_O_2_ 285.0503, found 285.0510.

*2-Phenyl-4-(4-fluorophenyl)-3-methyl-1,2,4-oxadiazolidin-5-one* (**3l**). Eluent cyclohexane/ethyl acetate 4: 1; yellow sticky oil (yield 70%). νmax/cm^−1^: 1730 (C=O). ^1^H-NMR (500 MHz, CDCl_3_): δ7.70 (d, *J*=7.2 Hz, 2H), 7.35–7.20 (m, 7H), 5.0 (q, *J*=9.5 Hz, 1H), 1.30 (d, *J*=9.5 Hz, 3H). ^13^C-NMR (500 MHz, CDCl_3_): δ 162.9, 153.3, 150.0, 134.7, 128.5, 127.1, 126.5, 123.2, 116.6, 115.7, 81.2, 19.2.HRMS-EI (*m*/*z*) calcd for C_15_H_13_ FN_2_O_2_ 271.0243, found 271.0250.

### 3.3. p38 MAP Kinase Assay

The p38 MAP kinase in vitro assays to determine the biological activity of the synthesized compounds were performed according to the published procedure [41]: test compounds were assayed in concentrations ranging from 10^−4^ to 10^−8^ M.Results are given as IC_50_values (µM): the IC_50_ values were measured by testing four concentrations of compounds at least 6-fold.

## 4. Conclusions

We have synthesized a series of azastilbenes where an isoxazolidine unit is the linker of the two vicinal 4-fluorophenyl/4-pyridine rings, with the C=C double bond of stilbene replaced by a *cis* locked C-N bond.

The synthesized compounds have been evaluated as p38α inhibitors. The inhibition data show that the two best inhibitors, 3e and 3f, strongly reduce the activity of p38α (80 nM and 150 nM, respectively). These data suggest that the core ring system of the reported 4-fluoro-phenyl/pyridinyl inhibitors can be replaced with the structurally simple C-N bond of the 5-membered ring and still obtain good inhibitory capacity.

## Figures and Tables

**Figure 1 molecules-26-01745-f001:**
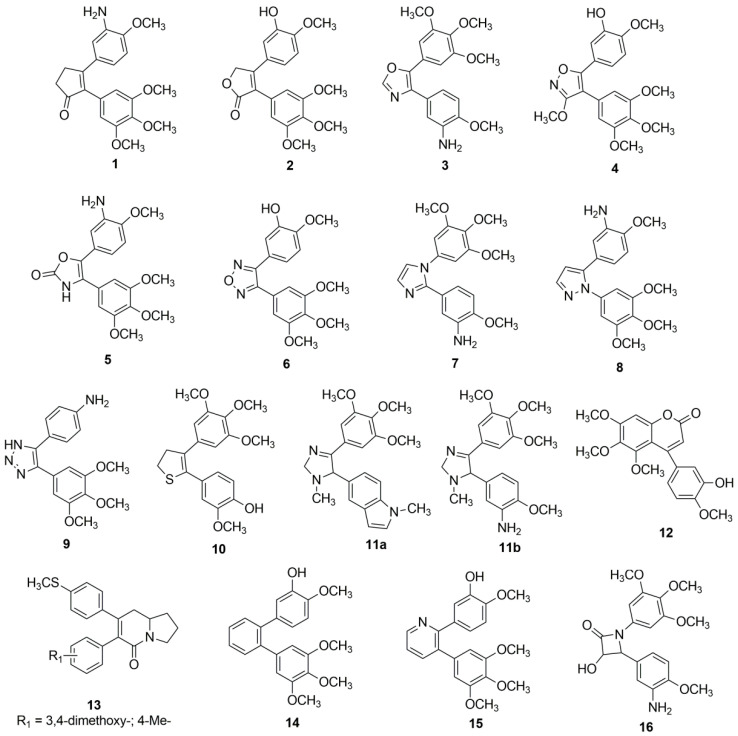
Selected *cis*-restricted stilbene-like analogs endowed with biological activity.

**Figure 2 molecules-26-01745-f002:**
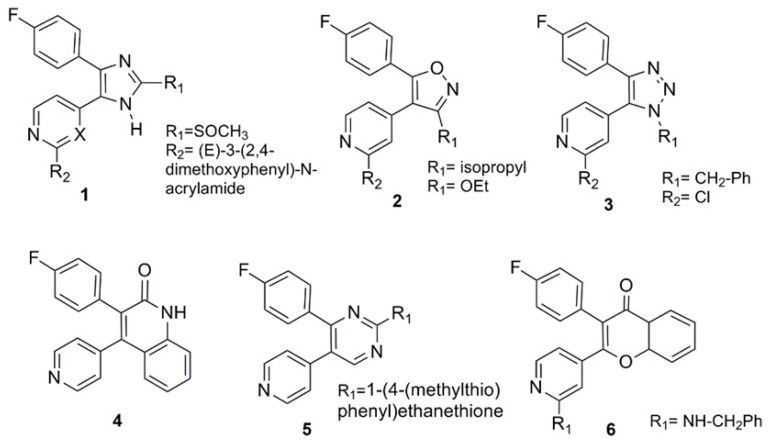
Stilbene-based p38 inhibitors characterized by a vicinal 4-fluorophenyl/4-pyridine motif.

**Figure 3 molecules-26-01745-f003:**
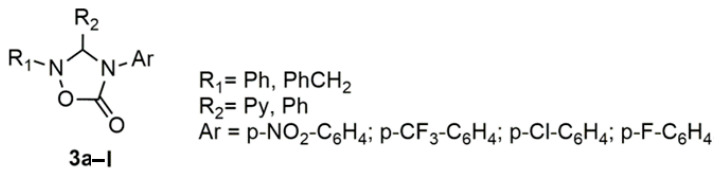
*Cis*-restricted azastilbenes.

**Figure 4 molecules-26-01745-f004:**
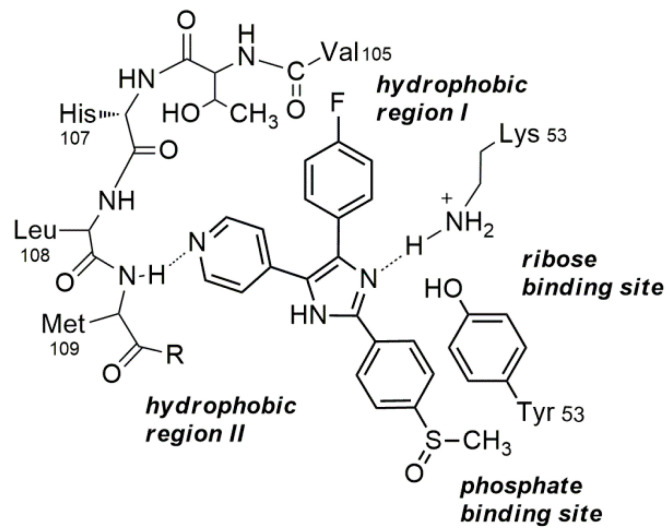
Binding mode of SB 203,580 at the p38α MAP kinase ATP binding pocket.

**Figure 5 molecules-26-01745-f005:**
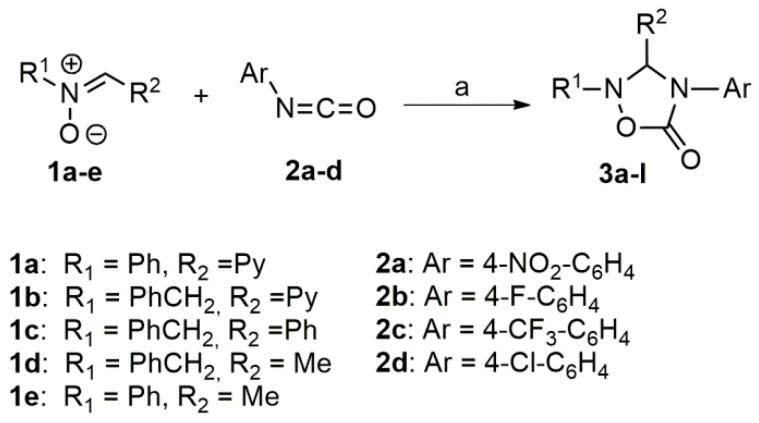
Reaction of nitrones **1a**–**e** with isocyanates **2a**–**d**. Reagents and conditions: (a) anhydrous acetone, room temperature, 5 h, 58–78% yields.

**Figure 6 molecules-26-01745-f006:**
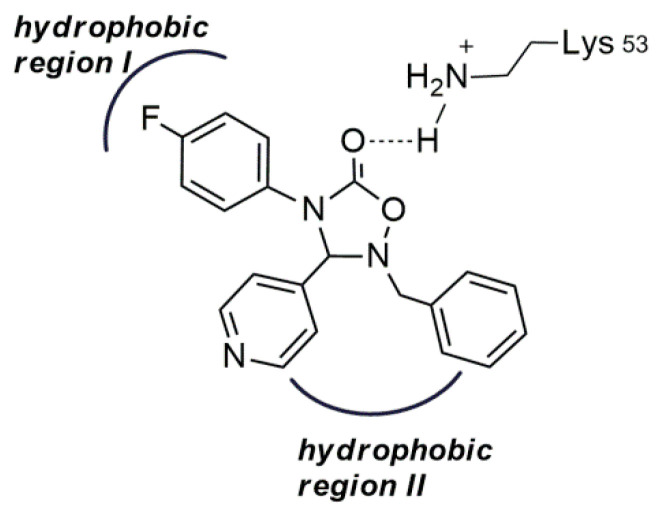
Possible binding mode of **3e** to the ATP site of p38α.

**Table 1 molecules-26-01745-t001:** Synthesis of 1,2,4-oxazolidinyl-5-ones **3a–l** by 1,3-dipolar cycloaddition.

Entry	R^1 a^	R^2 a^	Ar	Product (Yield%)
1	Ph	Py	4-NO_2_-C_6_H_4_	3a (76%)
2	Ph	Py	4-F-C_6_H_4_	3b (77%)
3	Ph	Py	4-CF_3_-C_6_H_4_	3c (75%)
4	Ph	Py	4-Cl-C_6_H_4_	3d (78%)
5	PhCH_2_	Py	4-F-C6H4	3e (62%)
6	PhCH_2_	Py	4-NO_2_-C_6_H_4_	3f (58%)
7	PhCH_2_	Ph	4-F-C_6_H_4_	3g (60%)
8	PhCH_2_	Ph	4-NO_2_-C_6_H_4_	3h (55%)
9	PhCH_2_	Me	4-F-C_6_H_4_	3i (75%)
10	Ph	Me	4-F-C_6_H_4_	3l (78%)

^a^ Substituents at R^1^ and R^2^ positions: Ph = phenyl;Py = pyridyl; Me = methyl.

**Table 2 molecules-26-01745-t002:** IC_50_values for synthesized compounds **3a–l** in p38α MAP kinase inhibition.

Entry	Compound	R^1^	R^2^	Ar	IC_50_-Value (μM) ^a^
1	3a	Ph	Py	4-NO_2_-C_6_H_4_	2.1
2	3b	Ph	Py	4-F-C_6_H_4_	0.1
3	3c	Ph	Py	4-CF_3_-C_6_H_4_	2.0
4	3d	Ph	Py	4-Cl-C_6_H_4_	1.5
5	3e	PhCH_2_	Py	4-F-C_6_H_4_	0.08
6	3f	PhCH_2_	Py	4-NO_2_-C_6_H_4_	0.15
7	3g	PhCH_2_	Ph	4-F-C_6_H_4_	15.1
8	3h	PhCH_2_	Ph	4-NO_2_-C_6_H_4_	60.5
9	3i	PhCH_2_	Me	4-F-C_6_H_4_	18.5
10	3l	Ph	Me	4-F-C_6_H_4_	22.8
	SB 203580				0.3

^a^ p38α MAP kinase activity was determined by the formulation of Ultra Glo Promega kit assay (Promega Italia srl, Milan, Italy). Data shown are mean ± S.D. of four separated experiments.

## Data Availability

Not applicable.

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
