# Peer review of "Synthesis and Biological Evaluation of 2,3,4-Triaryl-1,2,4-oxadiazol-5-ones as p38 MAPK Inhibitors"

_molecules, 2021, doi:10.3390/molecules26061745_

Round 1
Reviewer 1 Report
This paper is likely to be of interest to Molecules readers, but the authors should consider correction of the following suggestions:
Major:
- Copies of NMR spectra are missing. Please provide them in Supplement.
- Please provide data of binding mode and docking results of reference compound. There is a lack of discussion or comparison of synthesized derivatives and lead (reference) compound along results, conclusion sections. It is hard to value presented resunts.
Minor:
- Table 2. Reference compound with test results should be added to the table. Additionally, please insert suitable discussion regarding comparison of reference and studied compounds in text.
- Please revise and correct Reference section since there are many small discrepancies in format of citations
- According, to latest Molecules format Conclusions should be placed at the end of manuscript.
Author Response
This paper is likely to be of interest to Molecules readers, but the authors should consider correction of the following suggestions:
Major:
- Copies of NMR spectra are missing. Please provide them in Supplement.
We thank the Reviewer for the time devoted in revising the manuscript and for his valuable suggestions. The copies of NMR spectra have been added in the Supplementary Material.
- Please provide data of binding mode and docking results of reference compound. There is a lack of discussion or comparison of synthesized derivatives and lead (reference) compound along results, conclusion sections. It is hard to value presented resunts.
We agree with the Reviewer in considering the importance of docking studies to rationalize the reported biological data. However, in this study, we have hypothesized a possible binding mode of 3e with the ATP site of p38α, on the basis of literature data regarding docking investigations on imidazole-based inhibitor SB203580 with the crystal structures of p38α MAPK (see reference [37], in the manuscript). In this 2D study we reported that the location of residues important for binding with the enzyme and for kinase activity of reference compound 3e completely fit with the catalytic site of the enzyme. Complete docking studies of the more active compounds and of other p38 MAPK inhibitors, will be of course performed in future ad hoc studies. In order to better clarity this issue, we have re-written some sentences at lines 128, 194-195 and 206 of the manuscript.
Minor:
- Table 2. Reference compound with test results should be added to the table. Additionally, please insert suitable discussion regarding comparison of reference and studied compounds in text.
We thank the reviewer for having highlighted this missing data. We have added the test result obtained from compound SB 203580 in Table 2. We have also inserted a brief discussion about the comparison of this data with those obtained from the synthesized compounds at lines 182-185.
- Please revise and correct Reference section since there are many small discrepancies in format of citations
We have revised the Reference section according to Molecules format.
- According, to latest Molecules format Conclusions should be placed at the end of manuscript.
The Conclusion section is placed at the end of the manuscript. It is after the “Experimental Section” and before “Supplementary Materials”.
Reviewer 2 Report
The authors describe the synthesis of a series of azastilbene derivatives that were prepared as novel potential inhibitors of p38 MAPK. They conducted biological assays and molecular modeling to determine that several of the synthesized compounds showed inhibitory activity towards the kinase. The descriptions of the experimental and theoretical methods are thorough and well explained. The manuscript is clear and well written. It can be published after the authors fix some grammatical errors in the text.
Author Response
The authors describe the synthesis of a series of azastilbene derivatives that were prepared as novel potential inhibitors of p38 MAPK. They conducted biological assays and molecular modeling to determine that several of the synthesized compounds showed inhibitory activity towards the kinase. The descriptions of the experimental and theoretical methods are thorough and well explained. The manuscript is clear and well written. It can be published after the authors fix some grammatical errors in the text.
We thank the Reviewer for the time devoted in revising the manuscript and for the positive comments. We have checked the manuscript and corrected the grammatical errors.
Reviewer 3 Report
The study provide the Synthesis of 2,3,4-triaryl-1,2,4-oxadiazol-5-ones as a inhibitors of p38 MAPK. The major contribution is to replace the C=C double bond by a conformationally restricted C-N bond.
The author give details about the system, the synthesis, and the measurement of inhibition. The result is consist with the data provided.
Minor,
1. Page 6, line 202. "The docking result", but there's no docking described in the study, so other word should be used here.
2. It's better to give annotations for Ph,Py,Me under table 1
Author Response
The study provide the Synthesis of 2,3,4-triaryl-1,2,4-oxadiazol-5-ones as a inhibitors of p38 MAPK. The major contribution is to replace the C=C double bond by a conformationally restricted C-N bond. The author give details about the system, the synthesis, and the measurement of inhibition. The result is consist with the data provided.
Minor,
- Page 6, line 202. "The docking result", but there's no docking described in the study, so other word should be used here.
We thank the Reviewer for the time devoted in revising the manuscript and for his valuable comments. We completely agree in considering that the term “docking” was improperly written since we investigated the possible binding mode of the synthesized molecole on the basis of literature data. Thus, we have substituted “docking result” with “possible binding mode” at lines 194 and 206.
- It's better to give annotations for Ph, Py, Me under table 1
The annotation of Ph, Py and Me substituents were placed under Table 1.
Round 2
Reviewer 1 Report
Suitable corrections were made. I accept this manunuscript in present form.